# CRISPR/Cas9 Mutagenesis through Introducing a Nanoparticle Complex Made of a Cationic Polymer and Nucleic Acids into Maize Protoplasts

**DOI:** 10.3390/ijms242216137

**Published:** 2023-11-09

**Authors:** Bettina Nagy, Ayşegül Öktem, Györgyi Ferenc, Ditta Ungor, Aladina Kalac, Ildikó Kelemen-Valkony, Elfrieda Fodor, István Nagy, Dénes Dudits, Ferhan Ayaydin

**Affiliations:** 1Agribiotechnology and Precision Breeding for Food Security National Laboratory, Institute of Plant Biology, HUN-REN Biological Research Centre, 6726 Szeged, Hungary; nagy.bettina@brc.hu (B.N.); dudits.denes@brc.hu (D.D.); ayaydin.ferhan@brc.hu (F.A.); 2Laboratory of Cellular Imaging, Core Facilities, HUN-REN Biological Research Centre, 6726 Szeged, Hungarykelemen.ildiko@brc.hu (I.K.-V.); 3Department of Medical Microbiology, University Medical Center, University of Groningen, 9700 Groningen, The Netherlands; 4Department of Physical Chemistry and Materials Science, University of Szeged, 6720 Szeged, Hungary; ungord@chem.u-szeged.hu; 5Institute of Biochemistry, HUN-REN Biological Research Centre, 6726 Szeged, Hungary; fodor.elfrieda@brc.hu (E.F.); nagyi@baygen.hu (I.N.); 6SeqOmics Biotechnology Ltd., 6782 Mórahalom, Hungary; 7Functional Cell Biology and Immunology Advanced Core Facility (FCBI), Hungarian Centre of Excellence for Molecular Medicine (HCEMM), 6720 Szeged, Hungary; 8Faculty of Medicine, Albert Szent-Györgyi Health Centre, Interdisciplinary R&D and Innovation Centre of Excellence, University of Szeged, 6725 Szeged, Hungary

**Keywords:** poly (2-hydroxypropylene imine), nanoparticle, polyplexes, DNA oligonucleotide, plasmid uptake, maize protoplasts, GFP expression, DsRed fluorescent marker protein, mutagenesis, CRISPR/Cas9

## Abstract

Presently, targeted gene mutagenesis attracts increasing attention both in plant research and crop improvement. In these approaches, successes are largely dependent on the efficiency of the delivery of gene editing components into plant cells. Here, we report the optimization of the cationic polymer poly(2-hydroxypropylene imine) (PHPI)-mediated delivery of plasmid DNAs, or single-stranded oligonucleotides labelled with Cyanine3 (Cy3) or 6-Carboxyfluorescein (6-FAM)-fluorescent dyes into maize protoplasts. Co-delivery of the GFP-expressing plasmid and the Cy3-conjugated oligonucleotides has resulted in the cytoplasmic and nuclear accumulation of the green fluorescent protein and a preferential nuclear localization of oligonucleotides. We show the application of nanoparticle complexes, i.e., “polyplexes” that comprise cationic polymers and nucleic acids, for CRISPR/Cas9 editing of maize cells. Knocking out the functional *EGFP* gene in transgenic maize protoplasts was achieved through the co-delivery of plasmids encoding components of the editing factors Cas9 (pFGC-pcoCas9) and gRNA (pZmU3-gRNA) after complexing with a cationic polymer (PHPI). Several edited microcalli were identified based on the lack of a GFP fluorescence signal. Multi-base and single-base deletions in the *EGFP* gene were confirmed using Sanger sequencing. The presented results support the use of the PHPI cationic polymer in plant protoplast-mediated genome editing approaches.

## 1. Introduction

Maize is one of the most indispensable crops worldwide. It is also a preferred model in plant research focused on the development of genome editing tools for gene-specific mutagenesis. Significant progress has already been achieved in the improvement of maize traits through utilizing CRISPR/Cas9 editing technology (see review by Jiang et al. [1]). A key step in using the CRISPR/Cas9 system is the delivery of plasmid constructs into plant cells or intact tissues using a variety of methods (see review by [2]). As an alternative, synthetic short DNA molecules (30–50 mer oligonucleotides) were successfully used for oligonucleotide-directed mutagenesis (ODM) in plants [3,4,5,6]. Membrane-bound plant protoplasts serve as ideal editing objectives, not only for the transient evaluation of editing reagents [7] but also for the production of desired mutants in those systems in which plants can be regenerated from protoplast-derived cultured cells [8].

The introduction of plasmids encoding the CRISPR/Cas9 editing system into plant protoplasts is frequently based on polyethylene glycol (PEG) treatment and has been applied using a variety of plant species [7,8,9,10,11,12]. Considering the advantages and the disadvantages of this methodology, increasing attention is being devoted to various nanoparticle technologies as delivery methods in the genome editing of plants (see review by Vats et al. [13]). Recently, Mahmoud et al. [14] published an improved protocol for plasmid encapsulation using Lipofectamine™ LTX with PLUS Reagent (Thermo Fisher Scientific, Waltham, MA, USA) to induce donor DNA delivery in sweet orange protoplasts with low cytotoxic effects on cells. Positively charged cationic polymers are extensively used for nucleic acid delivery systems into mammalian cultured cells [15]. These nanoparticles are typically made of cationic monomers that contain primary, secondary, and/or tertiary amine groups that can be complexed with negatively charged nucleic acids to form condensed polymer–nucleic acid nanoparticle complexes called “polyplexes”. In the case of the optimal mixing ratio, the charge of the complex remains positive, which makes it possible to internalize into cells based on its interaction with the negatively charged cell membranes. Polyethylenimine (PEI) and poly (2-dimethylamino) ethylmethacrylate (pDMAEMA) are two classes of cationic polymers, which both have been used for DNA delivery into mammalian cells [1,16]. Nucleic acid dissociation from polyplexes can occur before or after nuclear entry, consequently protecting nucleic acids from degradation [17]. It was shown that polyplex migration towards the nucleus is not through random diffusion. Rather, they are transported on microtubules, which results in polyplex accumulation in the perinuclear region [18]. Thus, this provides cationic polymers with an advantage over cationic lipids since cationic polymers enhance DNA delivery into the nucleus, unlike cationic lipids [19].

Regarding the application of cationic polymer–nucleic acid nanoparticle complexes, (polyplexes) in plant transformation, Finiuk et al. [20] reported that pDMAEMA-based cationic polymers could transport plasmid DNA harboring the *EYFP* yellow fluorescent protein gene sequence into tobacco protoplasts. An et al. [21] used condensed nanostructures obtained from the combination of plasmid GFP and cationic polymers, such as pDMAEMA or PEI, for incubation and consequent plasmid introduction into *Arabidopsis thaliana* and *Nicotiana benthamiana* protoplasts or leaves. After plasmid uptake, the transient expression of the marker gene was demonstrated through monitoring the GFP fluorescence. Encouraged by the successful use of cationic polymers for plasmid DNA uptake into plant cells, in the present work, we have tested a novel polymer, namely, poly (2-hydroxy-propylene imine) (PHPI), for introducing plasmids as well as oligonucleotides into maize protoplasts. Previously, this cationic polymer was reported to yield a high transfection efficiency in HeLa cells [22]. After synthesis of the PHPI molecules, several parameters of plasmid delivery into maize protoplasts were optimized to achieve highly efficient delivery. Furthermore, plasmids of the CRISPR/Cas9 editing system were complexed with PHPI-based nanoparticles and introduced into protoplasts isolated from the transgenic, EGFP-expressing maize suspension culture. Microcalli lacking fluorescent signals were identified in these cultures. Sequencing of *EGFP* gene fragments from these tissues has revealed deletions or the insertion of nucleotides in this marker gene. The presented experiments support the use of this methodology for targeted gene mutagenesis in plant systems.

## 2. Results and Discussion

### 2.1. Synthesis and Molecular Weight Determination of the Cationic Polymer (Poly(2-hydroxy-propylene Imine), PHPI)

Since positively charged cationic polymers can interact with negatively charged DNA via electrostatic interactions, poly (2-(*N*,*N*-dimethylamino) ethyl methacrylate) (pDMAEMA) and polyethylenimine (PEI) have been widely used for DNA delivery in animal cells (see review by Bono et al. [23]). Based on its lower toxicity, pDMAEMA has considerable potential for DNA delivery into plant cells [20,21]. In the present work, we have tested an additional cationic polymer, poly (2-hydroxypropylene imine) (PHPI), as a new alternative for nucleic acid delivery into plant protoplasts. Following the protocol by Zaliauskiene and coworkers [22], we have synthesized this cationic polymer with slight modifications.

In order to determine the molecular weight of the synthesized polymer, the refractive index (Figure 1A) was measured for different concentrations of the polymer dispersion, ranging from 1 to 5 mg/mL. Based on the Zimm equation and the Debye plot (Figure 1B), the molecular weight (Mw) of the polymer was determined to be 66.4 ± 0.6 kDa using SLS measurements. The N/P ratio, indicating the ratio of positively chargeable polymer amine (N = nitrogen) groups to negatively charged nucleic acid phosphate (P) groups, is possibly one of the most important physicochemical properties of the polymer-based gene delivery vehicles [24]. Knowing the average Mw of the polymer, during the formation of DNA/polymer complexes, a N/P ratio of 9.2 was found to be optimal when 1 μg of DNA (plasmid/oligonucleotide) is mixed with 2 µg of polymer.

### 2.2. The Concentration of the Cationic Polymer and the Plasmid/Polymer Ratio Influence the Efficiency of the Delivery of GFP Plasmids

After polymer synthesis, the obtained polymer was stored as a dry powder following dialysis against water and lyophilization. For experiments, the dry lyophilizate was dissolved in sterile water at the final concentration of 10 mg/mL. As a high concentration of polymer decreased the viability of protoplasts, a concentration series was prepared from polymer stock solution and tested to determine the polymer quantity that provides the highest delivery efficiency of 1 µg of plasmid.

Keeping the amount of plasmid at a constant value, the resulting polyplexes carried different charges. It is also critical for polyplexes to have a net positive charge in order to interact with the negatively charged cell membrane, which can be achieved through adjusting the polymer-to-nucleic-acid ratio. Cationic polymer molecules can be neutralized by nucleic acids if an inadequate amount of polymer is used.

As can be seen on Figure 2, the highest delivery efficiency of 1 µg of plasmid was achieved using 2 µg of PHPI.

### 2.3. Incubation Time with the Complex Determines the Efficiency of the Delivery of GFP Plasmids

The experimental protocol used allowed us to analyze the effect of the incubation time of polyplexes with protoplasts on the yield of GFP-positive maize protoplasts. Based on the delivery efficiency data obtained in these experiments, incubation of protoplasts with polyplexes for 30 min ensured the highest average number of GFP-positive maize protoplasts (1.7%) (Figure 3). Statistical significance analyses showed that incubation times between 15 and 60 min can be used with comparable efficiency. This finding differs from the Arabidopsis protoplast infection results, when a significantly longer incubation time was applied [21].

As shown in Figure 4A,B, the incubation of maize protoplasts with polyplexes composed of plasmid DNA (pGFP, [25]) vector containing the *GFP* gene and the cationic polymer generated a considerable number of green fluorescent cells. When culturing the protoplast-derived cell population in culture media (N6M), EGFP-positive cells could divide and form multicellular colonies (Figure 4B). In a representative experiment with the optimized conditions, up to 2000 GFP-expressing maize protoplasts could be detected in a population of 15,000 protoplasts after treatment with polyplexes with 1 µg of plasmid DNA and 2 µg of PHPI cationic polymer in a volume of 200 µL. This corresponds to up to a 13% delivery efficiency in one well. In this case, using the same plasmid and polymer ratio as in experiments shown in Figure 2 and Figure 3, the delivery efficiency was much higher. These differences may be explained due to more optimal starting conditions of the recipient protoplasts in the experiments presented in Figure 4.

This optimizing study used a fine cell suspension of H1233 maize genotypes lacking the plant regeneration potential. Therefore, follow-up experiments were based on another suspension culture system (SZ17, Mórocz) possessing shoot regeneration capability.

### 2.4. Cationic Polymer Poly (2-hydroxypropylene Imine) (PHPI) Can Be Complexed with Short Synthetic DNA Molecules and Used for the Introduction of Oligonucleotides into the Membranes and Nuclei of Maize Protoplasts

Synthetic oligonucleotide-directed mutagenesis (ODM) attracts increasing attention as a targeted genome editing approach in plants [4]. Especially in light of the GMO debate, it is important to note that oligonucleotides are synthesized according to the sequence of a target gene with only a single nucleotide alteration. Consequently, there is no foreign gene introduction; therefore, plants with a point-mutated gene resulting from ODM have been altered in a way that can occur naturally. In ODM, the introduction of oligonucleotides into plant cells is a key step. Here, we tested the use of polyplexes for the uptake of fluorescently labeled DNA molecules into the nuclei of maize protoplasts. When we generated polyplexes containing 1 µg of FAM-labeled oligonucleotide and 2 µg of polymer, the delivery efficiency was high (Figure 5A). In attempts to ensure optimal viability of treated protoplasts, we found that the reduction of the uptake-inducing components (0.5 µg oligonucleotide with 1 µg polymer) can have positive effects, especially on the nuclear accumulation of oligonucleotides. However, the Cy3-labelled oligonucleotides were also detectable in outer membranes (Figure 5B).

### 2.5. Co-Delivery of Synthetic Oligonucleotides (ssODNs) and Plasmids into Maize Protoplasts through a PHPI Cationic Polymer

Sauer et al. [26] reported that simultaneous delivery of ssODNs and CRISPR-Cas9 plasmids into flax protoplasts resulted in a much higher targeted genome editing frequency compared with treatment using only the CRISPR/Cas9 components. In the cited study, PEG was used to introduce ssODNs and plasmid molecules into protoplasts. We tested whether the PHPI cationic polymer-based transformation could also be applied to the co-delivery of oligonucleotide and plasmid molecules. For visualization and tracking purposes, the mixture of GFP-expressing plasmids (green) and Cy3 fluorescently labelled oligonucleotides (red) was used for these uptake experiments. As shown by pictures in Figure 6, this methodology can ensure co-delivery. Interestingly, a differential accumulation pattern can be seen, as the green fluorescent protein is localized in the cytoplasm and in the nuclei, while oligonucleotides are preferentially found in the nuclei.

### 2.6. The PHPI Cationic Polymer-Induced Uptake of CRISPR/Cas9 Plasmids Can Generate Deletion and Insertion Mutations in the Marker GFP Transgene

Since the use of the CRISPR/Cas9 system for plant genome editing requires the co-expression of a generic Cas9 endonuclease and one or more specific single guide RNAs, the efficiency of this multicomponent system largely depends on the means of delivery of responsible macromolecules into plant cells or organs [7,27]. The above-described experiments and results have demonstrated the successful use of a novel cationic polymer (PHPI) for the introduction of functional plasmids into maize cells. On this basis, we have generated a reliable experimental system with the proper markers to test the CRISPR/Cas9 editing of the transgenic *EGFP* gene. In this approach, protoplasts with a green fluorescent signal were exposed to plasmids encoding the editing factors Cas9 (pFGC-pcoCas9) and gRNA (pZmU3-gRNA) in the presence of a cationic polymer. The later plasmid also carried the red fluorescent protein marker gene (*DsRed*), and the red fluorescent cells were counted as transformed cells. We used fluorescence microscopy to identify and assess the phenotypical changes involving fluorescence emission with a margin of error due to the variation in fluorescence emissions. In four treatment experiments, the average frequency of DsRed-positive, transformed cells was determined to be 4.1% with variation between 1.1% and 6.2%. In the DsRed-expressing cells, the loss of GFP signal is an indication of a targeted mutation. In the present set of experiments, the average frequency of DsRed-positive cells lacking GFP signal was scored as 1.65% with variation between 1.2% and 2.0%.

During the subsequent culturing, the maize cells proliferated as an in vitro culture and formed microcalli. The identification of microcalli developed from plasmid-targeted protoplasts was ensured through detection of the expressed DsRed fluorescent protein as a marker (Figure 7(A,C2)). The loss of green fluorescent signals in the recipient transgenic protoplast-derived microcalli served as a phenotypic sign for mutation in the *EGFP* gene (Figure 7(B,C3)). For confirmation of targeted mutations at a genomic level, DNA samples were isolated from six microcalli tissues lacking green fluorescent signals. After amplification of the target region of the EGFP gene, Sanger sequencing revealed multi-base and single-base deletions. Examples are shown in Figure 7D. In clone-2, we could detect a nucleotide insertion that resulted in an inversion. Nucleotide insertions have been detected previously in other protoplast-based CRISPR/Cas9 editing studies after PEG-mediated plasmid uptake [8]. With regard to the canonical Cas9 cleavage site, three nucleotides upstream of the PAM site, the positions of the present mutations were located in the 5′ direction from the Protospacer Adjacent Motif (PAM) sequence: 5′-NGG-3′.

In this report, we present supportive evidence that in addition to the published cationic polymers, such as polyethylenimine (PEI) and poly(2-(*N*,*N*-dimethylamino) ethyl methacrylate) (pDMAEMA), a novel nanostructure, poly(2-hydroxypropylene imine) (PHPI), can introduce DNA molecules in plant protoplasts. Since this approach proved to be suitable for the simultaneous introduction of different types of DNAs, PHPI can be an efficient tool in CRISPR/Cas9 editing of plant genomes. In the present study, maize protoplasts served as recipients in DNA uptake. This type of application can be extended to the transfection of plant cells and tissues. The pDMAEMA type of cationic polymer was reported as a transfection agent for tobacco leaves [21]. Therefore, experiments can be proposed for the use of PHPI molecules for editing morphogenic maize cells in order to produce novel breeding stocks.

## 3. Materials and Methods

### 3.1. Synthesis and Quantification of the Cationic Polymer Poly(2-hydroxypropylene Imine) (PHPI)

The cationic polymer poly(2-hydroxypropylene imine) was in-house synthesized according to a slightly modified method published by Zaliauskiene et al. [22]. A mass of 1,3-diamino-2-propanol (1.5 g, 16.6 mmol) was dissolved in a larger volume of methanol (8 mL). The synthesized polymer was dialyzed against sterile ultrapure water and was subsequently lyophilized. The lyophilized polymer was dissolved in sterile double-distilled water at a 10 mg/mL concentration.

The molecular weight was determined using dynamic and static light scattering (DLS and SLS) using a MalvernZetasizer Nano ZS ZEN 4003 (Malvern, UK) apparatus equipped with a He-Ne laser (λ = 633 nm) at 25 ± 0.1 °C. The refractive index gradient was calculated using a Mettler Toledo Excellence RM50 refractometer (Columbus, OH, USA).

### 3.2. Theoretical Background of the Molecular Weight Measurement

For the determination of the average molecular weight, SLS measurement was used. Three parallel measurements were carried out and the determination was performed based on the Debye plot (KC/R as a function of the C concentration) using the Zimm Equation (1). The slope of the fitting gives the second virial coefficient (2A2) and the intercept gives the reciprocal of the molecular weight (1/Mw).
KC/R = 1/(M_w P) + 〖2A〗_2 C(1)
where K is a constant dependent on the sample dn/dc, C is the sample concentration in g/mL, R is the Rayleigh ratio (the ratio of scattered light intensity to incident light intensity), Mw is the average molecular weight, and A2 is the second virial coefficient. P is an angular dependent term (by default, assumed to be 1 for small molecules <15 nm radius).

### 3.3. Isolation and Culture of Maize Protoplasts

*Zea mays* L. (H1233) callus cells grown in suspension culture provided by Sándor Mórocz (Cereal Research Institute, Szeged, Hungary) were cultured in N6M medium in the presence of 0.5 mg/L 2,4-dichlorophenoxyacetic acid (2,4-D) and used for protoplast isolation as described by Mórocz et al. [28]. For the knockout experiments, protoplasts were isolated from a transgenic B71 cell suspension. These cells were generated through biolistic bombardment of pEGAD plasmid (Accession: AF218816.1) to produce the Enhanced Green Fluorescent Protein (EGFP)-expressing maize cells.

### 3.4. Introduction of Plasmid DNAs and Synthetic Oligonucleotides into Maize Protoplasts through a Cationic Polymer

Since the primary aim of the present study was to optimize some key components of polyplex-based uptake protocols, the Results and Discussion section already provides some details of methodology. Plasmid DNA vectors pGFP [25], containing the *GFP* gene, or pEGAD were used as a reporter gene in these transfection experiments. First, the plasmid DNA was dissolved in 25 µL “R-medium” with 0.75 M mannitol. The R-medium contained: 1 mM CaCl_2_. 2H_2_O, 4 mM MgSO_4_. 7H_2_O, 620 mM KH_2_PO_4_, and 3.07 mM MES at pH 5.7. After addition of the cationic polymer, the polyplex mixture was diluted to 200 µL volume with R medium with 0.75 M mannitol. The polyplexes were added to the maize protoplasts in a dense droplet placed at the bottom of one of wells of an 8-well LabTek II chambered cover glass (Thermo Scientific, Waltham, MA, USA, #155409). After 30 min of incubation (or various incubation times as shown in Figure 3), 250 µL of protoplast culture medium (ppN6M, see [28]) was added onto wells. After incubation at room temperature for 30 min, the supernatant was removed and fresh ppN6M was added to the wells. This step was repeated twice.

As an alternative approach, we generated polyplexes with Cy3- and 6-Carboxyfluorescein (6-FAM)-conjugated fluorescent single-stranded oligonucleotides. The chemical synthesis of oligonucleotide molecules was performed using a DNA/RNA/LNA H-16 synthesizer (K&A Laborgeraete, Schaafheim, Germany) using standard β-cyanoethyl phosphoramidite chemistry at a nominal scale of 0.2 μmol. The oligonucleotides were purified via reverse phase HPLC followed by cation exchange using Dowex 50 resin (St. Louis, MO, USA).

### 3.5. Cationic Polymer Complexed with CRISPR/Cas9 Editing Plasmids for Targeted Mutagenesis of the Transgenic EGFP Gene in Cultured Maize Cells

Protoplasts were isolated from a suspension culture of stable transgenic maize cells expressing the Enhanced Green Fluorescent Protein (EGFP, pEGAD vector) transgene under the control of the 35S viral promoter. For knockout of this *EGFP* gene, 0.5 µg of a plasmid expressing both Cas9 (pFGC-pcoCas9, 13,330 bp, plasmid Addgene: 52,256) and gRNA (pZmU3-gRNA plasmid, 4820 bp, Addgene: 53,061) was complexed with 2 µg of PHPI cationic polymer. The gRNA (GGGCGAGGAGCTGTTCACCG_GGG) was designed by Benchling software (Benchling [Biology Software]. (2020). Retrieved from https://benchling.com) and it was cloned into the pZmU3-gRNA vector. For the safe identification of the GFP-minus transgenic cells, a red fluorescent protein marker gene (*DsRed* monomer derived from the 35S-DsRed monomer-NosT plasmid, Addgene: 79,183) was inserted into the same gRNA vector. The 35S promoter and the DsRed fragment were amplified with cloning primers with XhoI and EcoRI restriction enzymes sites: XhoI Fw: 5′-ccg ctcgag gagggctattgagacttttc-3′ and EcoRI Rev: 5′-cg gaattc gatctagtaacatagatgac-3′. The amplified and digested DsRed fragment was ligated into the same enzyme-digested pZmU3-gRNA vector. After the PHPI cationic polymer-induced uptake of editing plasmids, the treated protoplasts were cultured in ppN6M medium, where the maize cells could form multicellular microcalli.

### 3.6. PCR Amplification and Sequencing of the Green Enhanced Fluorescence Transgene

For analysis of CRISPR-mediated nucleotide editing, PCR amplifications were performed with the following *EGFP* gene-specific primers: GFP_Forward: 5′-ccaaccacgtcttcaaagca-3′ and GFP_Reverse: 5′-cggtggtgcagatgaacttc-3′. Phusion Hot Start II High Fidelity DNA Polymerase (Thermo Scientific) was used for the PCR reactions with the following cycle conditions: 1. Initial denaturation: 98 °C for 3 min. 2. Denaturation: 98 °C for 10 s. 3. Annealing: 67 °C for 30 s. 4. Extension: 72 °C for 20 s. (2–4 steps: 30 cycles) 5. Final extension: 72 °C for 10 min (Appendix A). Amplified fragments were purified using a GeneJet Gel Extraction kit (Thermo Scientific) according to the manufacturer’s recommendation. Fragments of 317 nucleotides were used for Sanger sequencing.

### 3.7. Confocal Fluorescence Microscopy

Olympus Fluoview FV1000 (Olympus Life Science Europa GmbH, Hamburg, Germany), Leica SP5 (Leica Microsystems GmbH, Wetzlar, Germany) confocal laser scanning microscopes and a Visitron spinning disk microscope were used for imaging. A 488 nm excitation laser was used for GFP and 543 nm laser was used for Cy3. GFP emission was detected between 505 and 530 nm, and Cy3 emission was detected between 570 and 670 nm. Composite images were prepared using ImageJ software (National Institutes of Health, Bethesda, MD, USA, version 1.41).

### 3.8. Statistical Analyses

Statistical analyses were performed using Student’s *t*-tests.

## Figures and Tables

**Figure 1 ijms-24-16137-f001:**
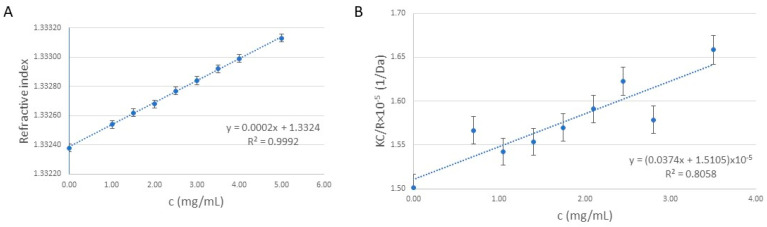
(**A**) The refractive index of the synthesized polymer samples as a function of the concentration. (**B**) The Debye plot used to estimate the molecular weight of the polymer.

**Figure 2 ijms-24-16137-f002:**
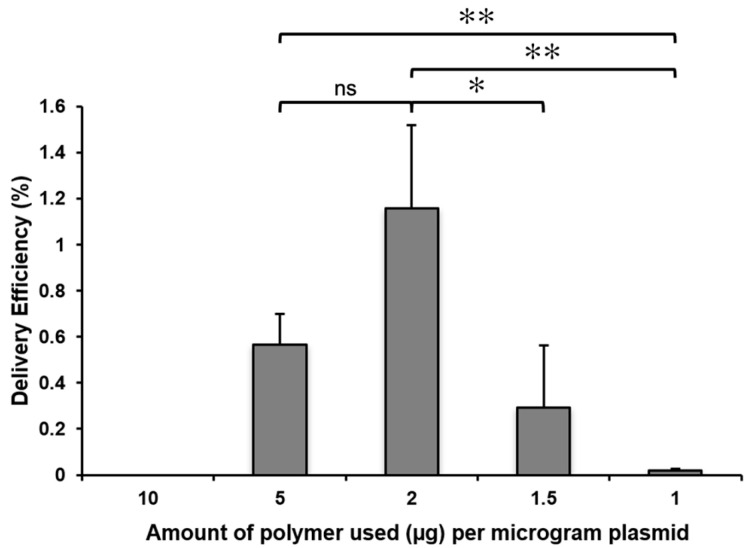
Delivery efficiencies of the pGFP plasmid into maize protoplasts depends on the polymer concentration. Maize protoplasts were treated with polyplexes formed using 1 µg pGFP plasmid and cationic polymer (PHPI) in the indicated quantity. Delivery efficiencies were analyzed 24 h after the polyplex treatment through counting the number of GFP expressing protoplasts under confocal laser scanning microscopy. Graph shows mean values (±SD) from three separate experiments. Significance was tested using a Student’s *t*-test (ns: non-significant, *: 0.01 < *p* < 0.05, **: *p* < 0.01).

**Figure 3 ijms-24-16137-f003:**
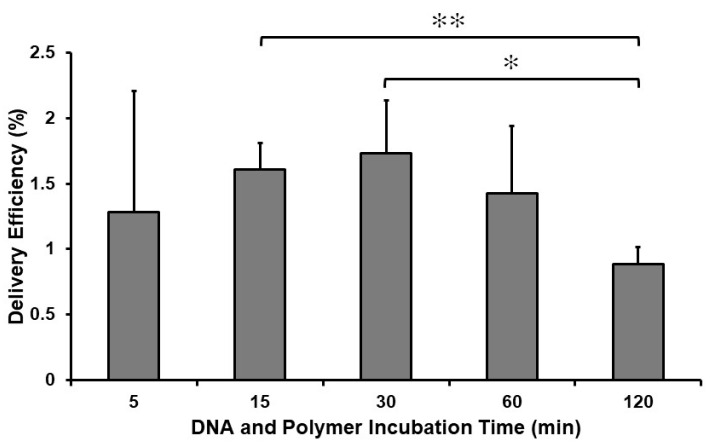
Maize protoplasts were incubated with polyplexes formed using 1 µg GFP plasmid and 2 µg cationic polymer for the indicated durations. At the end of these incubation periods, supernatants were removed and fresh protoplast media (ppN6M) were added. Delivery efficiencies were analyzed 24 h after the polyplex treatment through counting the number of GFP-expressing protoplasts under confocal laser scanning microscopy. Graph shows mean values (±SD) from three separate experiments. Significance was tested using a Student’s *t*-test (*: 0.01 < *p* < 0.05, **: *p* < 0.01).

**Figure 4 ijms-24-16137-f004:**
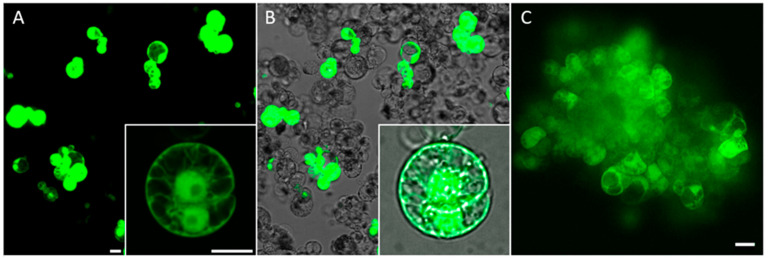
(**A**) Detection of green maize protoplasts indicates the uptake of the pGFP plasmid. (**B**) Merged image of brightfield and fluorescence is shown 24 h after treatment. Insets show a dividing cell after 5 days post treatment. In this experiment, 3 µL PCV of maize protoplasts was treated with polyplexes formed of 1 µg pGFP plasmid and 2 µg cationic polymer. After 1 d or 5 d (inset) following the polyplex treatment, GFP-expressing protoplasts were visualized under confocal laser scanning microscopy. (**C**) An example of the formation of a multicellular colony from GFP-positive protoplasts as an indication of a stable transformation event induced through polyplex treatment. The picture was taken after two weeks of culture in ppN6M protoplast-culturing medium. Scale bars are 25 µm.

**Figure 5 ijms-24-16137-f005:**
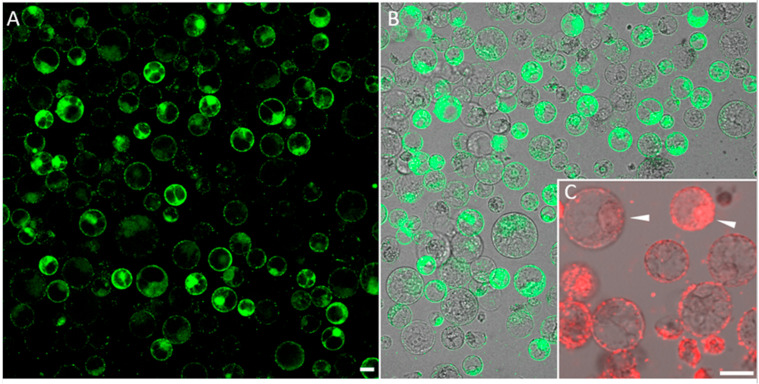
(**A**) Detection of the FAM signals in membranes and nuclei of the maize protoplasts after treatment with polyplexes formed using 1 μg oligonucleotide and 2 µg cationic polymer (PHPI) in a highly dense protoplast culture. (**B**) Brightfield image superimposed onto a fluorescence image to indicate the total number of protoplasts in the same field. (**C**) Inset shows the accumulation of oligonucleotides in membranes and nuclei of maize protoplasts after treatment with polyplexes formed using 0.5 µg Cy3-labeled oligonucleotide and 1 µg cationic polymer (PHPI). Arrowheads highlight two protoplasts with Cy3 accumulation in the nuclei. Scale bars are 25 µm.

**Figure 6 ijms-24-16137-f006:**
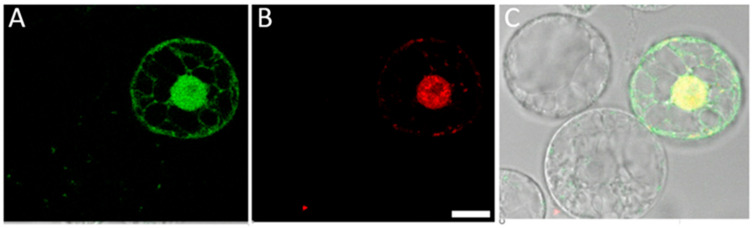
Cationic polymer PHPI can co-deliver both plasmid and oligonucleotide molecules into maize protoplasts. (**A**) GFP-expressing protoplast 1 day after polyplex treatment. (**B**) Delivery and retention of Cy3-labeled oligonucleotides and their nuclear accumulation. (**C**) Brightfield-merged green and red fluorescence images indicating successful co-delivery of both the plasmid and the oligonucleotide to the same protoplast. Maize protoplasts were treated with polyplexes formed using 0.5 µg GFP plasmid, 0.5 µg Cy3-labeled oligonucleotide, and 2 µg cationic polymer. Protoplasts were visualized under confocal laser scanning microscopy 24 h after the polyplex treatment. Scale bar is 10 µm.

**Figure 7 ijms-24-16137-f007:**
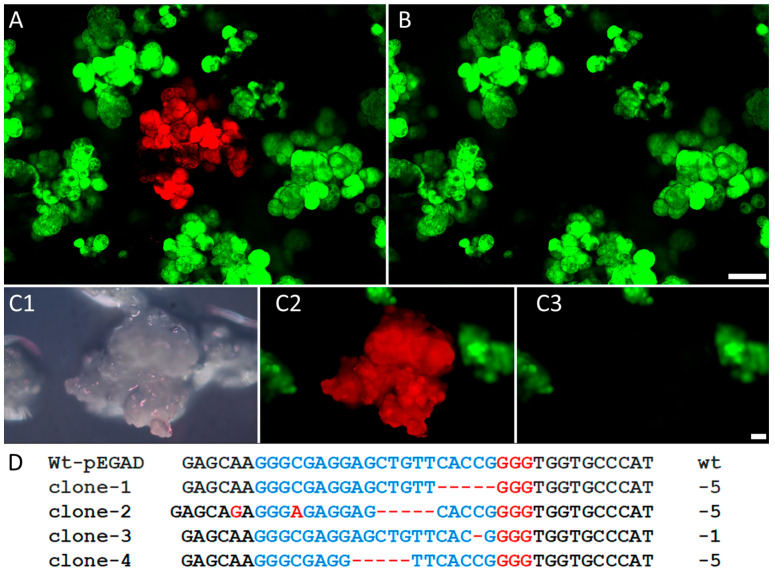
Phenotypic identification of maize microcalli with a knockout *EGFP* gene after the introduction of CRISPR/Cas9 plasmid vectors through cationic polymer (PHPI) treatment of transgenic maize protoplasts carrying the *EGFP* gene. A half microgram of each of the plasmids expressing Cas9 (pFGC-pcoCas9) and gRNA (pZmU3-gRNA) was complexed with 2 µg PHPI cationic polymer. (**A**) Protoplast-derived microcalli expressing EGFP (mother culture) and DsRed (transformed/edited) imaged using confocal microscopy. (**B**) The same microcalli population that shows a lack of green signal for the DsRed-positive callus. (**C1**) White light epi-illumination image of maize protoplast-derived microcalli under stereo dissection microscopy. (**C2**) Fluorescence image of the same microcalli expressing the DsRed marker gene showing the origin of these cells from a protoplast carrying the modified pZmU3-gRNA plasmid introduced via cationic polymer treatment. (**C3**) The same field of microcalli that shows an absence of green signal for the corresponding red callus. (**D**) Multi-base deletion and single-base deletion in the *EGFP* gene confirmed using Sanger sequencing of the *EGFP* fragment from knockout microcalli. Red GGG: PAM sequences; blue letters: CRISPR target sequence; black letters: wild type (wt) sequences, red dash lines: deleted nucleotides; red letters: inserted nucleotides/mismatch.

## Data Availability

Data are contained within the article.

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
