# Peer review of "CRISPR/Cas9 Mutagenesis through Introducing a Nanoparticle Complex Made of a Cationic Polymer and Nucleic Acids into Maize Protoplasts"

_ijms, 2023, doi:10.3390/ijms242216137_

Round 1
Reviewer 1 Report
Comments and Suggestions for Authors
Dear authors,
I consider that the work presents a novel technique that could improve the transfection of protoplasts and their subsequent regeneration. It is known that plant regeneration is the bottleneck of this technique, so it would be very important to present the efficiency of plant regeneration and editing after applying this transfection technique.
L58 add references for the use of protoplasts in other species (González et al., 2020; Liu et al., 2022; Najafi et al., 2022; etc)
Review the scale of the y-axes in graph 1
Statistical analysis is missing in figure 2
In Figure 3 they claim that 30 min was the most efficient time but in the legend of the figure they comment that there were no significant differences. Statistics should be reviewed to ensure that 30 min is the best incubation time.
L194 it is not correct to make this statement because it depends on the regulations of each country. The insertion of a certain amount of bases can be considered a GMO. Add references.
It would have been interesting to report the percentage of regenerated and edited plants.
L357 check repeat ¨0.5-0.5 µg plasmids¨
Best regards.
Author Response
Dear Reviewer,
Thank you very much for your review. We appreciate your suggestions. Please find our answers below and the corrections can be found in the revised manuscript.
Q1: L58 add references for the use of protoplasts in other species (González et al., 2020; Liu et al., 2022; Najafi et al., 2022; etc)
A1: We added the suggested references and Lin et al. 2022, too.
Q2.: Review the scale of the y-axes in graph 1
A2: As it can be seen on Fig.1 A, the y-axis shows the changes in the refractive indexes of the polymer solutions in different concentrations. While the molecular weight measurements require diluted solutions, the measured refractive indexes are close to the pure water: only the last two digits are altered.
On Figure 1B, the values of the y axis also became more visible after the correction.
Q3.: Statistical analysis is missing in figure 2
A3: We added the statistical analysis to the figure 2.
Q4: In Figure 3 they claim that 30 min was the most efficient time but in the legend of the figure they comment that there were no significant differences. Statistics should be reviewed to ensure that 30 min is the best incubation time.
A4: We added the statistical analysis to the figure 3, too.
We added a sentence: ˝Statistical significance analyses showed that incubation times between 15 to 60 minutes can be used with comparable efficiency.˝
Q5: L194 it is not correct to make this statement because it depends on the regulations of each country. The insertion of a certain amount of bases can be considered a GMO. Add references.
A5: We reworded this chapter, which hopefully resulted in a more precise version.
Q6: It would have been interesting to report the percentage of regenerated and edited plants.
A6: We agree with you, but unfortunately the maize line used in the presented experiments is not a regenerable genotype.
Q7: L357 check repeat ¨0.5-0.5 µg plasmids¨
A7: corrected.
We hope that revision helps in understanding our results.
Best regards,
Györgyi Ferenc
Reviewer 2 Report
Comments and Suggestions for Authors
Reviewer comments:
The manuscript entitled “CRISPR/Cas9 mutagenesis by introducing nanoparticle complex of cationic polymer and nucleic acid into maize protoplasts.” by Nagy et al. I found this topic interesting, demonstrates a polyplex-based uptake protocols in Maize protoplasts. But I have few concerns related to the research article. I am asking authors to revise the manuscript carefully considering my comments for possible publication in “International Journal of Molecular Sciences”.
I have given my comments.
• The present investigation will be a good contribution to the genetic improvement of Maize
using nanoparticle complex of cationic polymer.
• Line No 302: The authors requested to check and correct ‘1,5g 1,3-diamino-2-propanol’.
• Line No 340: The authors requested to check and correct the sentence ‘polyplex mixture was diluted to 200 µL with R medium’.
• Line No 343: Authors, please provide the incubation time ‘At the end of this incubation’.
• Line No 357: Authors must check and correct the sentence ‘0.5-0.5 µg plasmids’
• Authors should mention the number of gene edited and co-transformation percentage.
• The authors are requested to include PCR confirmation gel image.
The submitted manuscript may be acceptable for publication after a major revision.
Comments on the Quality of English LanguageMinor editing is required in Materials and methods.
Author Response
Dear Reviewer,
Thank you very much for your review. We appreciate your suggestions. Please find our answers below and the corrections can be found in the revised manuscript.
Q1: Line No 302: The authors requested to check and correct ‘1,5g 1,3-diamino-2-propanol’.
A1: we have corrected
Q2: Line No 340: The authors requested to check and correct the sentence ‘polyplex mixture was diluted to 200 µL with R medium’.
A2: corrected
Q3: Line No 343: Authors, please provide the incubation time ‘At the end of this incubation’.
A3: corrected
Q4: Line No 357: Authors must check and correct the sentence ‘0.5-0.5 µg plasmids’
A4: corrected
Q5: Authors should mention the number of gene edited and co-transformation percentage.
A5: You can find in the following part: ˝ In four treatment experiments, the average frequency of DsRed positive, transformed cells was determined to be 4.1% with variation between 1.1% to 6.2%. In the DsRed expressing cells, the loss of GFP signal is an indication for targeted mutation. In the present set of experiments, the average frequency of DsRed positive cells lacking GFP signal was scored as 1.65% with variation between 1.2% to 2.0%.˝
Q6: The authors are requested to include PCR confirmation gel image.
A6: We included it in a supplementary document.
Comments on the Quality of English Language
Q7: Minor editing is required in Materials and methods.
A7: Materials and methods section is edited.
We hope that revision helps in understanding our results.
Best regards,
Györgyi Ferenc